# Predictors of Development of Hepatorenal Syndrome in Hospitalized Cirrhotic Patients with Acute Kidney Injury

**DOI:** 10.3390/jcm10235621

**Published:** 2021-11-29

**Authors:** Roula Sasso, Ahmad Abou Yassine, Liliane Deeb

**Affiliations:** Department of Gastroenterology, Staten Island University Hospital—Northwell Health, New York, NY 10305, USA; aabouyassine1@northwell.edu (A.A.Y.); LDeeb1@northwell.edu (L.D.)

**Keywords:** predictors, hepatorenal syndrome, cirrhosis, mortality, acute kidney injury

## Abstract

Hepatorenal syndrome (HRS) is a type of acute kidney injury (AKI), occurring in patients with decompensated liver cirrhosis and is associated with high mortality. We aim to describe the predictors associated with the development of HRS in cirrhotic patients with AKI. We retrospectively analyzed 529 cirrhotic patient encounters with AKI across all Northwell Health institutions between 1 January 2015 and 31 December 2018. We performed multivariate analyses to determine independent predictors of development of HRS. Alcoholic cirrhosis was the most common identified etiology of cirrhosis. The mean Model for End-Stage Liver Disease Scorewas18 (±7). Ascites was the most commonly identified clinical feature of portal hypertension. Infection was identified in 38.4% of patients with urinary tract infection/pyelonephritis being the most common. Spontaneous bacterial peritonitis occurred in 5.9% of patients. The most common cause of AKI was pre-renal. Hepatorenal syndrome was identified in 9.8% of patient encounters. Predictors of HRS were history of ascites, serum creatinine >2.5 mg/dL, albumin <3 g/dL, bilirubin >2 mg/dL and spontaneous bacterial peritonitis. We demonstrate strong predictors for the development of HRS which can aid clinicians to attain an early diagnosis of HRS, leading to prompt and targeted management and improving outcomes.

## 1. Introduction

Acute kidney injury (AKI) is a common complication in patients with decompensated liver cirrhosis with the most common cause being dehydration and volume depletion [1,2]. AKI in cirrhotic patients is associated with high morbidity and mortality [3], with an estimated medial survival of less than 50% at 3 months [4,5,6].

Mechanisms of development of AKI in cirrhotic patients are variable. Cirrhotic patients are at risk of decreased effective arterial blood volume secondary to splanchnic vasodilatation, resulting in decreased renal perfusion. Cirrhotic patients are also at increased risk of acute or chronic gastrointestinal bleeding, as well as volume depletion due to medications such as lactulose and diuretics, which can result in further reduction of their effective arterial volume.

Hepatorenal syndrome (HRS) is a form of AKI, occurring in patients with decompensated liver cirrhosis. The definition of HRS has evolved over the past several years. The updated definition of HRS by the International Club of Ascites(ICA) in 2015 [7] renamed the previously known HRS-type 1 as HRS-AKI, and abandoned the previously required doubling of serum creatinine to >2.5 mg/dL within 2 weeks, to replace it with the definition of AKI based on the updated KDIGO guidelines which is an increase in serum creatinine by 0.3 mg/dL within 48 h, or 1.5 times increase in baseline creatinine which is known or presumed to have occurred within the prior 7 days [8].

The pathophysiology of HRS-AKI is multifactorial and mostly attributed to an uncompensated hyperdynamic circulatory system, renal vasoconstriction and systematic inflammation [9,10,11]. The diagnosis of HRS-AKI is often demanding and complex due to the clinical challenge of differentiating between HRS-AKI and other causes of AKI (such as pre-renal azotemia and acute tubular necrosis (ATN)). In addition, it is required to exclude structural kidney and bladder diseases which in turn often results in delaying timely management.

Early diagnosis and treatment of hepatorenal syndrome is important as better prognosis substantially depends on timely management in this group of patients [12,13]. Treatment with albumin and terlipressin or vasopressors has clearly been shown to improve mortality [14,15,16]. In our study herein, we hypothesize that the risk of development of HRS-AKI can be predicted based on patient baseline clinical characteristics and laboratory values at the time of development of acute kidney injury. We therefore aimed to describe the variables associated with the development of HRS-AKI in cirrhotic patients with acute kidney injury to guide clinicians in determining the risk of development of HRS-AKI which would help attaining an early diagnosis by increasing clinical awareness specifically in this group of patients.

## 2. Materials and Methods

### 2.1. Study Population and Design

This was a retrospective case-control study of cirrhotic inpatients visits admitted across all Northwell institutes between 1 January 2015 and 31 December 2018. Patients below the age of 18 and those with outpatient hospital encounters were excluded. We identified cirrhotic patients using our institution’s Information Technology data warehouse; to identify patients with cirrhosis, we used the International Classification of Disease, ninth revision clinical modification (ICD-9-CM) codes as previously defined [17]. The corresponding ICD-10 codes (K70.30, K70.31, K74.6, K74.60, K74.5, K74.69) were selected to identify cirrhotic patients with medical encounters occurring after 1 October 2015. We then used the following ICD-9-CM codes to identify patient with AKI on admission: 584.5, 584.6, 584.7, 584.8 or 584.9 as previously defined [18] with the following corresponding ICD-10 codes: N17.0, N17.2, N17.8, N17.9. This study was approved by the Feinstein Institute for Medical Research, Northwell Health institutional review board.

### 2.2. Data Collection and Definitions

Using our data warehouse, clinical variables including patient demographics, medical and surgical history, cirrhosis history (including etiology, history of esophageal varices, ascites, and hepatic encephalopathy), laboratory data, vital signs were abstracted at the time of admission. Ascites on admission was identified by either physical exam documentation or radiologic evidence of ascites. Charleson Comorbidity Index (CCI) score was calculated [19] and adjusted for liver cirrhosis by excluding liver disease as a comorbidity. Additionally, home medications and medications administered during hospital stay, hospital length of stay, and outcomes (such as requirement of intubation, hemodialysis, and hospital mortality) were obtained. Charts and progress notes were then manually reviewed by RS (author) and AAY (author) to determine the presence of HRS, etiology of AKI other than HRS and suspected or confirmed infection.

AKI was defined using the KDIGO criteria as an increase in serum creatinine by 0.3 mg/dL or more within 48 h or by 1.5 times baseline or more within the last 7 days, or a urine output of less than 0.5 mL/kg/h for 6 h [20]. We defined pre-renal AKI to include patients with AKI secondary to hypovolemia, systemic vasodilation, and increased renal vascular resistance (such as compressive ascites). Intrinsic AKI included patients with tubular, glomerular, interstitial, and vascular injury. Postrenal AKI included extrarenal and intrarenal obstruction [21]. AKI due to HRS and cardiorenal AKI were separately identified to allow for more detailed analysis for the purpose of this study. The diagnosis of type 1 cardiorenal syndrome phenotype (i.e., AKI resulting from acute coronary syndrome or acute heart failure) was obtained from manual chart review and was as defined by AKI as per the KDIGO criteria resulting due to underlying acute cardiac pathology as previously described [22].

The primary outcome was development and diagnosis of HRS-AKI during hospital stay. Hepatorenal syndrome was diagnosed using the previously defined criteria [8,23]. Secondary outcomes included the following: need for intubation, hemodialysis requirement, hospital length of stay and hospital mortality.

### 2.3. Statistical Analysis

Patient characteristics and outcomes were presented as mean ± standard deviation for continuous variables, and frequencies and percentages for categorical variables. Variables were compared between the two groups of patients (patients who developed HRS vs. those who did not) using the student’s *t*-test or Mann-Whitney U-test for the continuous variables and the Pearson’s chi-square test or fisher’s exact test for the categorical variables as appropriate. To calculate odds ratios, continuous variables were dichotomized and the odds ratios with corresponding 95% confidence intervals were calculated for all statistically significant variables. A creatinine cutoff value of 2.5 mg/dL was chosen as a reasonable midrange between the mean creatinine value of patients with chronic kidney disease (CKD), and those without CKD to avoid skewing the predictors of progression to HRS based on baseline creatinine value. An albumin cutoff of 2 g/dL was chosen to reflect hypoalbuminemia. A bilirubin cutoff of 2 mg/dL was chosen to reflect significant hyperbilirubinemia. An INR cutoff of 1.5 was chosen to reflect impaired liver synthetic function. A value was considered statistically significant at a two-tailed test *p*-value less than or equal to 0.05. All statistical analyses were performed using SPSS (version 25).

## 3. Results

### 3.1. Patient Demographics, Baseline Characteristics and Outcomes

A total of 15,425 cirrhotic patient encounters were identified between 1 January 2015 and 31 December 2018. Of these patient encounters, 529 met our inclusion criteria (Figure 1). The mean age of our study population was 65 years (±12 years), and the majority of patients were male (*n* = 330, 62.4%) and Caucasian (*n* = 296, 56%). The most commonly identified etiology of cirrhosis was alcoholic cirrhosis (*n* = 233, 44%) followed by non-alcoholic steatohepatitis (NASH) cirrhosis (130, 24.6%). Only 17.9% of the patients had baseline chronic kidney disease and the mean adjusted CCI score was 3 (±2). Ascites was the most commonly identified clinical feature of portal hypertension (*n* = 207, 39.1%) followed by hepatic encephalopathy, with 66.7% of the patients having clinical signs of portal hypertension (Table 1). A total of 10 patients were identified to have alcoholic hepatitis.

On admission, mean systolic and diastolic blood pressure was 121 (±26 mmHg) and 68 (±12 mmHg) respectively. Mean creatinine in patients with normal kidney function as baseline and patients with known chronic kidney disease at baseline was 1.9 (±1.2) mg/dL and 2.9 (±2.4) mg/dL respectively. MELD score on admission was 18 (±7) and MELD- sodium (MELD-Na) score was 19 (±8). Mean and standard deviations of laboratory data are demonstrated in Table 1. Confirmed infection was identified in 38.4% of patients with the most common infection being urinary tract infections (UTI)/pyelonephritis (*n* = 51, 9.6%). Spontaneous bacterial peritonitis (SBP) was identified in 5.9% of patients (Table 1). Of the 22 patients who were identified to have bacteremia, 14 had gram positive bacteremia and 6 patients had multidrug resistant organisms as the cause of the bacteremia.

The most common cause of AKI was pre-renal in nature (35.7%) followed by intrinsic AKI (10.4%). HRS-AKI was identified in 9.8% of patient encounters. A total of 77 (14.6%) and 17 (3.2%) patients required mechanical ventilation and hemodialysis, respectively. Mean hospital length of stay was 10 (±9) days and 11.2% of the patients had in hospital mortality (Table 1).

### 3.2. Correlation of Patient Characteristics and Outcomes for HRS-AKI vs. no HRS-AKI

When comparing patient characteristics and outcome between the two cohorts (HRS-AKI group vs. no HRS-AKI group), those who developed HRS-AKI were more likely to have alcoholic cirrhosis (59.6% vs. 42.3%), an established history of ascites (75% vs. 34.6%) or hepatic encephalopathy (61.5% vs. 32.5%), and clinical evidence of portal hypertension (90.4% vs. 64.2%). Interestingly, those who developed HRS-AKI were more likely to have ascites present on admission and were more likely to have grade 3 ascites (large volume ascites) compared to those who did not develop HRS-AKI. Only 10 patients with a history of ascites, but no apparent ascites on admission developed HRS-AKI. Additionally, patients who developed HRS were more likely to have lower hemoglobin, platelet count, sodium and albumin levels on admission as well as higher creatinine, bilirubin and INR. Patients with HRS were also noted to have significantly higher MELD score on admission (24 ± 7 vs. 17 ± 7) and were more likely to have an established diagnosis of SBP during hospitalization as demonstrated in Table 2.

Those who developed HRS were more likely to get intubated (25 vs. 13.4%), to require hemodialysis (13.5% vs. 2.1%) and to expire during hospital stay (40.4% vs. 8%). Diagnosis of HRS was not associated with older age, higher CCI score, low systolic or diastolic blood pressure, increased hospital length of stay or increase in all cause infection (Table 2).

### 3.3. Odds for Development of Hepatorenal-AKI

All variables that were statistically significantly different in the previous correlative analysis were used to calculate odds ratios for the development of HRS-AKI. The bivariate analysis demonstrated that bilirubin >2 mg/dL was the strongest predictor for development of HRS (OR: 9.5, SD: 5.1–17.7), followed by a history of ascites (OR: 5.7, SD: 2.9–10.9). Interestingly, all degrees of ascites on admission were predictors of development of HRS-AKI, however ascites grade 3 was the strongest predictor (Appendix A). To establish independent predictors and adjusted odds ratios (aOR) for diagnosis of HRS-AKI, we conducted a multivariate analysis which showed that only a history of ascites, baseline serum creatinine >2.5 mg/dL on admission, albumin <2 g/dL, bilirubin >2 mg/dL and spontaneous bacterial peritonitis were statistically significant predictors for development of HRS (Table 3). After stratifying patients based on the presence of chronic kidney disease (CKD) at baseline, a creatinine >2.5 mg/dL was only a predictor of progression to HRS in those without CKD (Appendix A).

## 4. Discussion

The diagnosis of HRS-AKI provides a challenge for clinicians. Prior to confirming the diagnosis of HRS, patients need to receive appropriate volume resuscitation for 48 h, withdrawal of diuretics and other causes of AKI must be excluded. This leads to delays in the diagnosis of HRS when it is the case, with resulting deferral of the appropriate treatment with albumin infusions, terlipressin or other vasoconstrictors. In our study we demonstrate that clinical and laboratory features in cirrhotic patients admitted with AKI may assist in predicting progression to HRS-AKI and potentially result in earlier diagnosis of hepatorenal syndrome. Independent variables that can predict development of HRS are history of ascites, serum creatinine >2.5 mg/dL, albumin <2 g/dL, bilirubin >2 mg/dL and spontaneous bacterial peritonitis, with the strongest predictor being bilirubin >2 mg/dL (aOR: 7.9) followed by an established history of ascites (aOR: 5.8).

Studies on predictors of development of HRS are scarce and most studies are limited by difficulties in establishing the diagnosis and changing conceptual understanding of the pathophysiology and definition of HRS-AKI. A study published in 1993 [24] demonstrated that low sodium levels, high plasma renin and absence of hepatomegaly to be independent predictors of the development of HRS in non-azotemic cirrhotic patients. However, the definitions of HRS and acute renal insufficiency has since evolved thus limiting the applicability of this study’s findings. Acute renal insufficiency. A more recent study showed that increased creatinine, bilirubin, MELD score and decreased serum sodium and albumin to be predictors of development on HRS in patients with alcoholic liver cirrhosis [25]. This study is limited by small sample size and low generalizability as only alcoholic cirrhotic patients were included. A recent randomized control trial evaluated triggers of development of HRS-1 (now known as HRS-AKI) vs. HRS-2 (as previously defined [7]) as a secondary outcome and demonstrated that the most common trigger for HRS-1 was unidentifiable triggers followed by paracentesis of >5 L in the last 4 weeks [26]. This study however utilizes the 2007 ICA diagnostic criteria for HRS which may result in an underdiagnosis of HRS-AKI given the requirement of serum creatinine to be ≥1.5 mg/dL

Interestingly, compared to prior studies [24,25,27] our study does not show hyponatremia to be an independent predictor of development of HRS-AKI after adjusting for other variables, despite patients with HRS-AKI having significantly lower sodium levels compared to those without HRS-AKI in bivariate analysis. This could be due to the fact that our study controlled for the presence of ascites in the multivariate analysis, unlike the previously mentioned studies. Given that development of ascites and hyponatremia are both secondary to splanchnic vasodilatation and activation of neurohumoral systems (such as renin-angiotensin, aldosterone and antidiuretic hormone pathways) [28], resulting in free water retention and perpetuation of ascites, thus we consider that controlling for the presence of ascites might eliminate the statistical significance of hyponatremia.

Bilirubin >2 mg/dL was shown to be a strong predictor of development of HRS-AKI in our study. Several animal studies have suggested that cholestasis and elevated bilirubin result in hypotension and decreased vasopressor response [29,30]. Additionally, an underrecognized entity termed cholemic nephropathy has been described [31] in patients with cirrhosis and elevated bilirubin, with studies suggesting a decreased expression of aquaporin-2-receptors in patients with elevated bilirubin [32] and some studies have demonstrated that patients with elevated bilirubin and HRS-AKI have less response to terlipressin and albumin [14], suggesting a possible pathophysiologic contribution of elevated bilirubin in the development of HRS.

Hypoalbuminemia appears to predict development of HRS-AKI in our study. Low serum albumin represents decreased liver function and hypoalbuminemia eventually leads to increased fluid third spacing, ascites and intravascular volume depletion resulting in activation of the renin angiotensin pathway and kidney vasoconstriction [15], consistent with the understood pathophysiology of development of HRS. A history of established ascites on presentation was also a predictor for development of HRS in our study. This is also supported by the overactivation of neurohormonal responses previously described [28,33] and is consistent with the requirement for ascites to be present in order to diagnose HRS. All our patients with HRS were eventually found to have ascites during their hospital stay.

Serum creatinine >2.5 mg/dL was found to be an independent predictor of HRS (aOR: 2.5), but this was only statistically significant in patients who did not have CKD at baseline. This could reflect the severity of renal dysfunction in patient with HRS, however this could also be due to the previously defined diagnosis of HRS, requiring a creatinine greater than 2.5 mg/dL. Additionally, it is worth noting that serum creatinine may not provide a reliable and consistent estimate for the severity of renal dysfunction in cirrhotic patients, as these patients tend to have reduced muscle mass and hepatic creatine production [34]. Thus, we feel that using serum creatinine as a marker for prediction of HRS needs further evaluation. Some studies have suggested that cystatin C may be a good predictor of development of HRS [35], however results are conflicting for the use of this parameter in the diagnosis of HRS-AKI/HRS type 1 [36].

Spontaneous bacterial peritonitis was shown to be a strong predictor for development of HRS and has been suggested to precipitate HRS-AKI. One study demonstrated that HRS-AKI occurred in up to 30% of patients with SBP [37], whilst another study showed a diminished improvement in renal function in patients with HRS if their SBP was unresolved compared to those whose infection was controlled [38]. This is consistent with the suspected hyperinflammatory state and elevated levels of IL-6 and TNF-alpha noted in patients with SBP, and thought to be associated with the development of HRS [39,40]. Interestingly, a study demonstrated that administration of norfloxacin for SBP primary prophylaxis decreased the development of HRS-AKI [41].

Interestingly, patients with cirrhosis are at increased risk of developing infections such as SBP, clostridium difficile and aspiration pneumonias possible due to increased intestinal bacterial translocation that results from portal hypertension and increased intestinal transient time [42], or due to a decreased innate pulmonary immunity [43]. One study suggested that changes in immunity and increased risk of infections might also be present in patients without advanced fibrosis or cirrhosis [44]. Infections in cirrhotic patients can precipitate a systemic inflammatory response which may be a precursor to the development of HRS-AKI. In fact, a study by Sole et al. [45] demonstrated that patients who developed HRS-AKI had higher levels of inflammatory markers and cytokine levels.

In our study, HRS-AKI occurred in 9.8% of cirrhotic patient encounters admitted for AKI. This is consistent with other reported studies which demonstrated the incidence of HRS among cirrhotic patients to be within this range [25,27]. Mortality in patients with HRS in our study was 40%, which is consistent with other studies showing higher mortality in patients with HRS-AKI compared to other causes of AKI [4,46].

We recognize the limitations of our study being mainly inherent to its observational and retrospective nature. This study relies on the accuracy of medical documentation and ICD coding to establish our inclusion criteria. However, the ICD coding for identifying cirrhosis patients and AKI has been previously validated and have high sensitivity and specificity in identifying patients with the diagnosis in question [17,18]. Furthermore, the use of ICD codes can often overlook subtle etiologies or causes of clinical decompensation that are not billed for. Despite the diagnosis of development of HRS-AKI being identified by the authors through manual chart reviews, we do acknowledge that the diagnosis of HRS-AKI is challenging and at times a diagnosis of exclusion. Additionally, this study was conducted across all Northwell Health sites, some of which are not liver transplant centers and may possibly be limited in their ability to diagnose and manage decompensated cirrhotic patients with HRS-AKI. However, data was obtained across 12 different sites, enhancing the generalizability of our study.

## 5. Conclusions

In this study, we demonstrate that a history of ascites, serum creatinine >2.5 mg/dL, albumin <2 g/dL, bilirubin >2 mg/dL, and spontaneous bacterial peritonitis are predictors for the development of HRS. Establishing potential predictors can aid clinicians to classify patients as high risk or low risk for the development of hepatorenal syndrome and may have a key role in expediting the diagnosis which in turn will lead to earlier targeted management and improved survival. Further studies are needed to verify our established predictors of HRS as well as prospective studies that will aid in providing better evidence on the relevance and clinical benefit of these established predictors.

## Figures and Tables

**Figure 1 jcm-10-05621-f001:**
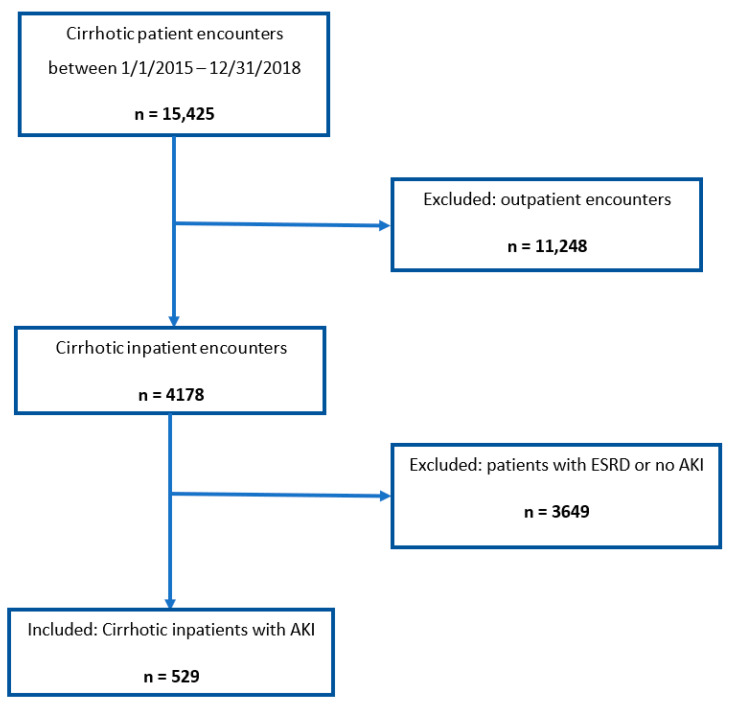
Consort diagram demonstrating inclusion and exclusion criteria.

**Table 1 jcm-10-05621-t001:** Clinical characteristics and outcomes of cirrhotic patients admitted with Acute Kidney Injury (AKI).

Variables	*n* = 529 *n* (%) or Mean (SD)
Age (years)	65 (12)
Gender (female)	199 (37.6%)
Race	
White/Caucasian	296 (56%)
African American/Black	83 (15.7%)
Asian	41 (7.8%)
Other	91 (17%)
Not specified/unavailable	18 (3.4%)
Cirrhosis etiology ^1^	
Alcoholic	233 (44%)
Non-alcoholic steatohepatitis	130 (24.6%)
Hepatitis C virus (HCV)	105 (19.6%)
Hepatitis B virus (HBV)	8 (1.5%)
Cryptogenic/other ^2^	66 (12.5%)
Charleson comorbidity index score	3 (2)
Chronic kidney disease	95 (17.9%)
History of ascites	207 (39.1%)
Ascites present on admission	183 (34.6%)
Ascites grade 1	80 (15.1%)
Ascites grade 2	64 (12.1%)
Ascities grade 3	39 (7.4%)
History of hepatic encephalopathy	187 (35.3%)
History of esophageal varices	176 (33.3%)
Clinical signs of portal hypertension	353 (66.7%)
Non-selective beta-blocker use during admission	169 (31.9%)
Vital signs	
Body mass index	29.7 (11.3)
Systolic blood pressure	121 (26)
Diastolic blood pressure	68 (12)
Hemoglobin (g/dL)	10.8 (2.4)
Platelet count (×10^3^)	160 (107)
Sodium (meq/L)	135 (6)
Creatinine (mg/dL)	2.1 (1.6)
Albumin (g/dL)	3.2 (0.6)
Bilirubin (mg/dL)	2.1 (3.9)
International normalized ratio (INR)	1.6 (1.0)
MELD score	18 (7)
MELD-Na score	19 (8)
Infection identified ^3^	203 (38.4%)
UTI/pyelonephritis	51 (9.6%)
Pneumonia/respiratory infections	36 (6.8%)
Cellulitis/other skin infections	30 (5.7%)
Spontaneous bacterial peritonitis	31 (5.9%)
Bacteremia	22 (4.2%)
Colitis/enteritis (other than clostridium difficile)	12 (2.3%)
Clostridium difficile	9 (1.7%)
Cholecytitis	7 (1.3%)
Other ^4^	13 (2.5%)
Cause of AKI	
Pre-renal	189 (35.7%)
Intrinsic	55 (10.4%)
Hepato-renal syndrome	52 (9.8%)
Cardiorenal	39 (7.4%)
Postrenal	10 (1.9%)
Unidentified	20 (3.8%)
Received Albumin	117 (22.1%)
Received octreotide	102 (19.3%)
Vasopressor requirement	89 (16.8%)
Mechanical ventilation requirement	77 (14.6%)
Hemodialysis requirements	17 (3.2%)
Hospital length of stay	10 (9)
Death	59 (11.2%)

^1^ Some patients may have more than one etiology of cirrhosis. ^2^ Includes cryptogenic cirrhosis, cardiac cirrhosis, primary biliary cholangitis, primary sclerosis cholangitis, autoimmune cirrhosis. ^3^ Some patients may have more than one identified infection. ^4^ Other infections included endocarditis, osteomyelitis, septic arthritis, perforated secondary peritonitis, and fungal infections.

**Table 2 jcm-10-05621-t002:** Variables associated with the development of HRS in cirrhotic patients admitted with Acute Kidney Injury.

Variables	Development of Hepato-Renal Syndrome		*p*-Value
	Yes (*n* = 52) *n* (%) or Mean (SD)	No (*n* = 477) *n* (%) or Mean (SD)	
Age (years)	63.2 (9.7)	65.5 (12.6)	0.191
Gender (female)	24 (36.2%)	175 (36.7%)	0.361
Race			
White/Caucasian	27 (52%)	269 (56.4%)	0.751
African American/Black	9 (17.3%)	74 (15.5%)	
Asian	4 (7.7%)	37 (7.8%)	
Other	9 (17.3%)	82 (17.2%)	
Not specified/unavailable	3 (5.8%)	15 (3.2%)	
Cirrhosis etiology			
Alcoholic	31 (59.6%)	202 (42.3%)	<0.05
Non-alcoholic steatohepatitis	7 (13.5%)	123 (25.8%)	0.05
Hepatitis C virus (HCV)	10 (19.2%)	95 (19.9%)	0.906
Hepatitis B virus (HBV)	2 (3.8%)	6 (1.3%)	0.146
Cryptogenic/other	3 (5.8%)	61 (12.8%)	0.141
Charleson comorbidity index score	3 (2)	3 (2)	0.206
Chronic kidney disease	5 (9.6%)	90 (18.9%)	0.099
History of ascites	39 (75.0%)	165 (34.6%)	<0.05
Ascities present on admission	42 (80.8%)	141 (29.6%)	<0.05
Ascities grade 1	7 (16.7%)	73 (51.8%)	<0.05
Ascities grade 2	18 (34.6%)	46 (9.6%)	
Ascities grade 3	17 (40.5%)	22 (15.6%)	
History of hepatic encephalopathy	32 (61.5%)	155 (32.5%)	<0.05
History of esophageal varices	17 (32.7%)	159 (33.3%)	0.926
Clinical signs of portal hypertension	47 (90.4%)	306 (64.2%)	<0.05
Non selective beta blocker use during hospitalization	12 (23.1%)	157 (32.9%)	0.149
Vital signs			
Body mass index	28.2 (6.7)	29.9 (11.7)	0.367
Systolic blood pressure	115 (26)	122 (26)	0.057
Diastolic blood pressure	64 (18)	68 (16)	0.059
Hemoglobin (g/dL)	10.0 (1.8)	10.9 (2.5)	<0.05
Platelet count (×10^3^)	129 (102)	164 (107)	<0.05
Sodium (meq/L)	131 (6)	136 (6)	<0.05
Creatinine (mg/dL)	2.7 (2.0)	2.0 (1.6)	<0.05
Albumin (g/dL)	2.8 (0.5)	3.2 (0.6)	<0.05
Bilirubin (mg/dL)	3.4 (4.0)	1.9 (3.9)	<0.05
International Normalized Ratio (INR)	1.9 (0.8)	1.6 (1.0)	<0.05
MELD score	24 (7)	17 (7)	<0.05
MELD-Na score	27 (7)	19 (7)	<0.05
Infection identified	25 (48.1%)	175 (36.8%)	0.113
UTI/pyelonephritis	7 (28.0%)	44 (25.3%)	0.771
Pneumonia/respiratory infections	1 (4.0%)	35 (20.1%)	0.050
Cellulitis/other skin infections	2 (8.0%)	28 (16.1%)	0.290
Spontaneous bacterial peritonitis	9 (36.0%)	18 (10.3%)	<0.05
Bacteremia	3 (12%)	19 (10.9%)	0.872
Colitis/enteritis	0 (0.0%)	12 (6.9%)	0.176
Clostridium difficile	2 (8.0%)	7 (4.0%)	0.371
Gallbladder infections	0 (0.0%)	7 (4.0%)	0.307
Other	2 (8.0%)	11 (6.3%)	0.751
Mechanical ventilation requirement	13 (25%)	64 (13.4%)	<0.05
Hemodialysis requirements	7 (13.5%)	10 (2.1%)	<0.05
Hospital length of stay	11.2 (7.6)	9.9 (8.8)	0.330
Death	21 (40.4%)	38 (8.0%)	<0.05

**Table 3 jcm-10-05621-t003:** Odds of development of HRS in a bivariate and multivariate analysis.

Variable	Bivariate Analysis		Multivariate Analysis	
	Crude OR (CI)	*p*-Value	Adjusted OR (CI)	*p*-Value
Alcoholic cirrhosis	2.0 (1.1–3.6)	<0.05	1.2 (0.5–2.4)	0.678
History of ascites	5.7 (2.9–10.9)	<0.05	5.8 (2.6–13.0)	<0.05
History of hepatic encephalopathy	3.3 (1.8–6.0)	<0.05	1.5 (0.7–3.1)	0.256
Hb < 11 g/dL	2.6 (1.4–5.0)	<0.05	1.7 (0.8–3.7)	0.160
Platelets < 150 (×10^3^)	2.4 (1.3–4.4)	<0.05	1.5 (0.7–3.2)	0.308
Sodium < 135 meq/L	3.1 (1.7–5.8)	<0.05	2.2 (0.9–4.7)	0.053
Cr > 2.5 mg/dL	3.0 (1.7–5.6)	<0.05	2.5 (1.2–5.5)	<0.05
Albumin < 2 g/dL	3.8 (1.5–9.5–8.0)	<0.05	3.9 (1.1–13.5)	<0.05
Bilirubin > 2 mg/dL	9.5 (5.1–17.7)	<0.05	7.9 (3.7–17.0)	<0.05
INR > 1.5	5.6 (2.9–11.1)	<0.05	1.4 (0.3–5.8)	0.630
Spontaneous bacterial peritonitis	4.9 (1.9–12.7)	<0.05	5.5 (1.4–12.2)	<0.05

## Data Availability

Data available in request.

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
