# Peer review of "Predictors of Development of Hepatorenal Syndrome in Hospitalized Cirrhotic Patients with Acute Kidney Injury"

_jcm, 2021, doi:10.3390/jcm10235621_

Round 1
Reviewer 1 Report
The hepatorenal syndrome is one of myriad causes of acute kidney injury in patients with acute or chronic liver disease. In this study, the authors looked at potential predictors of development of HRS in cirrhotic patients. I did appreciate the study and commend the authors on their work.
Specific comments:
- Please change "Roula Sasso, MD1,*; Ahmad Abou Yassine, MD1 ; Liliane Deeb and MD1" to "Roula Sasso, MD1,*; Ahmad Abou Yassine, MD1 ; and Liliane Deeb MD1".
- As per the journal's guidelines, the abstract should be a total of 200 words maximum.
- Please provide the actual IRB study/approval number.
- Please change "... between 1/1/2015 and 12/31/2018" to "... between 1 Jan, 2015 and 31 Dec, 2018".
- "3.3. Figures, Tables and Schemes", there is no need to have a separate section for the CONSORT diagram.
- "... are strong predictors for the development of HRS" - how do you qualify this? SBP has a large 95% CI of 1.2-11.5, even after adjusting for potential confounders. At this point, it is also crucial to ask yourself, which variable is the most important?
- It is important to distinguish between crude ORs and adjusted ORs (aORs) in the text.
- Were there missing or incomplete data seeing that data were collected from 12 different sites?
- The diagnosis of the hepatorenal syndrome is one of exclusion, entertained only after other potential causes of acute or subacute kidney injury have been ruled out. As an example, both glomerulonephritis and vasculitis can occur in patients with liver disease and should be suspected in patients with an active urine sediment containing red cells and red cell and other casts. There is definitely some challenge and ambiguity in identifying and diagnosing HRS.
Reviewer 2 Report
The hepatorenal syndrome plays a significant role for nephrologists. Liver disease is quite prevalent in our hospitals, we are consulted for elevated plasma creatinines in such individuals, and nephrologists witness the tragedy of decompensation often. The hepatorenal syndrome is a diagnosis of exclusion. Criteria include an elevation of the plasma creatinine, acute/subacute timing, absence of other causes of kidney injury, no obstruction, minimal to no hematuria, minimal to no proteinuria, and lack of improvement in renal function with volume expansion and holding diuretics.
Author list: Line 4. Roula Sasso, Ahmad Abou Yassine, Liliane Deeb and MD? This needs to be corrected
Abstract. The abbreviations CCI and MELD need to be spelled out before 1st use.
Methods. This was retrospective, from January 2015 to December 2018. Out of 15425 patients, 529 met the inclusion (inpatient, with a definition of acute kidney injury by the (2012) KDIGO recommendations (with stage 1 starting at an elevation of plasma creatinine of at least 1.5 times the baseline or an increase of >/= 0.3 mg/dL — there are urine output definitions, too, but these were not alluded to in the methods).
Line 65. Based on ICD-9-CM codes. When was the transition to ICD10 codes?
Line 73. How was the Charlson Comorbidity Index adjusted for liver cirrhosis?
Results.
Line 142. A serum creatinine > 2.5 mg/dL was a predictor of hepatorenal syndrome diagnosis – is this a baseline serum creatinine of > 2.5 mg/dL? That would be worth stating.
Discussion.
Line 159. The sentence, “In our study we demonstrate that clinical and laboratory features… can assist in earlier diagnosis.” This is not what the study demonstrates. The study revealed that there are some variables that correlate with the eventual diagnosis of hepatorenal syndrome (ascites, bilirubin, INR – those listed in Table 3). The authors did not perform a prospective study complete with sensitivity/specificity receiver operator curves to then demonstrate that there was a tool to use to diagnose hepatorenal syndrome any earlier. (The authors cite a prospective trial, Janicko 2015, that did.) Therefore it cannot be said that these clinical and laboratory features actually assist in earlier diagnosis. This sentence should be changed, perhaps suggesting that the variables may be of use in developing something for an earlier diagnosis.
Line 170. Stating that a prior study was limited by an outdated definition of hepatorenal syndrome and acute ‘kidney injury’ is a bit disingenuous. It is a syndrome, and the definition evolved. And in the purest sense of the hepatorenal syndrome, using the term acute ‘kidney injury’ is a complete misnomer; the kidneys have zero injury, but are responding to the pathophysiology of liver disease. In this case ‘acute renal insufficiency’ is so much more accurate (and our diagnostic terminology devolved). I recommend changing that sentence.
Line 175. Only at this point are types 1 and 2 mentioned. The difference should be defined if they want to retain this sentence (although the present manuscript is clearly focused on type 1).
Line 178. Again, being so dismissive of a 2007 definition of hepatorenal syndrome comes off as flippant. These early definitions are the foundation of today’s criteria for the syndrome. What, exactly, are the weaknesses of the earlier criteria? If there are pivotal points, then state them clearly. Simply because a definition dates to 2007 doesn’t negate its utility in 2021.
Line 180. “compared to prior studies our study does not show hyponatremia to be an independent predictor…” The authors attribute this to controlling for the presence of ascites in the multivariate model. Table 3 shows that hyponatremia in the bivariate model was a predictor. Their results do not negate anything that we undertand about hepatorenal physiology despite not attaining statistical significance in the multivariate model. Therefore, is it possible that this was lost by over-fitting in the multivariate model? Or that the sample size for a multivariate model was too small and this was a type II error? One of the papers (Ginès et al 1993) listed essentially all the factors of this manuscript as having predictive value for hepatorenal syndrome and nicely demonstrated a dramatic reduction in free water clearance from enrollment to time of diagnosis with hepatorenal syndrome in their table 3 (P < 0.009). (The foundational studies for hepatorenal syndrome typically had this elegance. The discussion could be greatly improved by demonstrating a tad more reverence for previous work.)
Line 213. Yes, plasma creatinine is insensitive for estimating the true glomerular filtration rate in the population with liver disease. This is also true for using creatinine in contemporary estimated glomerular filtration rates (Skluzacek, Paul A., et al. "Prediction of GFR in liver transplant candidates." American journal of kidney diseases42.6 (2003): 1169-1176.)
Line 234. ICD-9 codes, by definition, should be valid for cirrhosis and acute ‘kidney injury,’ otherwise this study would be detecting billing fraud. The authors are correct that the study relied on accuracy of medical documentation. Studies limited to billing codes have a massive handicap that the subtle mechanisms responsible for multiple organ failures are difficult (if not impossible) to detect with such methods. Perhaps weaknesses of using such a method (and even within medical documentation) should be elucidated.
Table 1. Surprisingly, with an average age of 65 years and liver disease, (some with ascites), the Charlson Comorbidity index score mean was 3. (An age of 65 years equates to 2.5 points. Mild liver disease is 1 point. Ascites (even though 39% of the dataset) and varices (33%) entail portal hypertension (i.e., 3 points). What skewed the CCI to be so low on average? Perhaps a comment should be put in the discussion.
Overall, the importance of this submission is largely supportive of a great body of literature that has founded our understanding (and definition) of hepatorenal syndrome. It was well-written in general. However, the studies that pre-dated this work seem to be diminished by the tone in the Discussion. The quality of billing data and what information that can be derived from these, histories, and laboratory data are over-stated. Discovering something that would help us discern hepatorenal syndrome from other causes of elevated plasma creatinine is laudable.
2021-11-09
Reviewer 3 Report
I have read with interest, and very carefully this study where authors' aim was to identify potential risk factors for development of HRS in patients with liver cirrhosis. While this is an important question to ask, and certainly identification of these predictors might help us treat patients sooner than later, there are several important issues that I would like the authors to clarify. I will list only major concerns:
Introduction
1. In my opinion, stratification of severity of the liver disease, MELD-Na score would be more accurate than just MELD score.
2. Line 27- you should specify in % range of morbidity and mortality in different studies.
3. Line 42- renal vasoconstriction plays significant part in HRS development ( https://www.ncbi.nlm.nih.gov/pmc/articles/PMC2600320/)
4. In introduction the authors should add a paragraph explaining other complication that patient with liver cirrhosis and NAFLD have, specifically related to infection ( https://pubmed.ncbi.nlm.nih.gov/33977096/
Methodology
5. Why cut of for creatinine of 2.5 mg/dl was chosen? Have you separated patient with AKI who previously had normal renal function from those with CKD who had AKI on CKD? How was AKI on CKD diagnosed?
6. You should also describe how cardiorenal AKI was differentiated from HRS.
7. As you are aware majority of patients with cirrhosis ( depending on severity) will have hypoalbuminemia. How was a cut off of 3 g/dL chosen? In my opinion choosing lower value such as 1.5 or 2 g/dL might be of interest as value of 3 g/dL is too high.
8. From your study sample, have you separated patient who had alcoholic hepatitis? This group is particularly prone to develop AKI and have worse prognosis than similar patients without AH https://www.ncbi.nlm.nih.gov/pmc/articles/PMC3124879/
Results:
9. Authors describe that 18% of their patients had CKD patients- what was their baseline creatinine?
10. Ascites- it is very important to report if patients had ascites during the admission or had history of it that was successfully managed with diuretics etc. That would have implication on results - is just history of ascites or actual presence of it of prognostic significance. For me as a clinician it is of very important significance as I consider the risk factors you have mentioned.
11. Table 1- you mention that non alcoholic steatohepatitis was present in 130 patients. NASH or NAFLD? These are different entities with former being more severe so I would like to bring this towards authors attention. How was NASH diagnosed- invasive ( liver biopsy) or it was non invasive ( biomarkers, elastography etc)
12. Table 1 also reports bacteremia. It would be important to be more specific as not every bacteremia is equal- cirrhotic patient have higher incidence of MDRs and these have worse outcome ( https://pubmed.ncbi.nlm.nih.gov/30391380/)
13. Table 2- gallbladder infection is amateurish term- cholecystitis would be much more elegant.
14. The lines 180-187- authors have no proof of this. Furthermore mean sodium levels was 135 which is a normal value. This paragraph and statement need to be revised.
15. In statistical analysis- authors should consult with statistician to check which variables in univariate analysis are associated with HRS development, and subsequently they should include variables found to be statistically significant in multivariate analysis. From these variables ( found to be statistically significant in multivariate analysis) a predictive score would be useful to be developed so we have sensitivity and specificity of that particular score
16. Role of beta blockers and their risk for HRS development should be investigated
References
17. Some of the references are too obsolete. In general, references should be within 10 years from the publication date. Please review and update accordingly. In addition to references listed above you should read and cite the following in appropriate sections
https://www.ncbi.nlm.nih.gov/pmc/articles/PMC5332418/
https://www.ncbi.nlm.nih.gov/pmc/articles/PMC4663201/
https://www.ncbi.nlm.nih.gov/pmc/articles/PMC3390443/
https://www.gastrojournal.org/article/S0016-5085(14)00306-0/fulltext
https://pubmed.ncbi.nlm.nih.gov/32846202/
https://pubmed.ncbi.nlm.nih.gov/33848394/
Round 2
Reviewer 1 Report
Thank you for the revisions.
Author Response
The authors appreciate the feedback
Reviewer 3 Report
The authors have addressed semi-successfully some of my comments. While the manuscript has been improved , I have further comments before the manuscript can be consider for publication:
- In methodology section, before statistical analysis you should explain what you mentioned in the response letter to me. Describe how did you choose cut off values of creatinine, bilirubin, albumin etc.
- Additionally, how did you differentiate between cardiorenal and hepatorenal syndrome is important to mention in methodology as well. You can use with more details response that you provided in the reviewers letter. All these information are essential to be included in methodology
- To my previous comment number 10 regarding ascites. Authors could, although it is labor intensive, to review charts manually , as physical exam must be documented on each admission, and some of these patients even had US exam during the admission. If paracentesis was done during admission it would be obvious that ascites was poorly controlled. This is even more important as authors elaborate that they controlled for ascites in relation to hyponatremia. In my opinion this is the major discrepancy and should be addressed. Authors have two options here: a) review all charts and document ascites as present or absent; or b) acknowledge this shortcoming in limitations and eliminate hyponatremia discussion as it is obvious that you did not do precise analysis
- I disagree that infections in NAFLD and cirrhosis are not important for development of HRS. I strongly suggest additional paragraph in discussion section that would describe this very important connection
